# Structural basis for proficient oxidized ribonucleotide insertion in double strand break repair

Joonas A. Jamsen [1✉], Akira Sassa[2], Lalith Perera [1], David D. Shock[1], William A. Beard[1] & Samuel H. Wilson [1✉]

Reactive oxygen species (ROS) oxidize cellular nucleotide pools and cause double strand breaks (DSBs). Non-homologous end-joining (NHEJ) attaches broken chromosomal ends together in mammalian cells. Ribonucleotide insertion by DNA polymerase (pol) μ prepares breaks for end-joining and this is required for successful NHEJ in vivo. We previously showed that pol μ lacks discrimination against oxidized dGTP (8-oxo-dGTP), that can lead to mutagenesis, cancer, aging and human disease. Here we reveal the structural basis for proficient oxidized ribonucleotide (8-oxo-rGTP) incorporation during DSB repair by pol μ. Time-lapse crystallography snapshots of structural intermediates during nucleotide insertion along with computational simulations reveal substrate, metal and side chain dynamics, that allow oxidized ribonucleotides to escape polymerase discrimination checkpoints. Abundant nucleotide pools, combined with inefficient sanitization and repair, implicate pol μ mediated oxidized ribonucleotide insertion as an emerging source of widespread persistent mutagenesis and genomic instability.

---

[1] Genome Integrity and Structural Biology Laboratory, National Institute of Environmental Health Sciences, National Institutes of Health, Research Triangle Park, NC, USA. [2] Laboratory of Chromatin Metabolism and Epigenetics, Graduate School of Science, Chiba University, Chiba, Japan. ✉email: joonas.jamsen@nih.gov; wilson5@niehs.nih.gov

Exposure to reactive oxygen species (ROS) can oxidize cellular macromolecules causing DNA damage, such as chromosomal single and double strand breaks (DSBs)[1,2]. Left unrepaired, DSBs may result in genome rearrangements leading to cancer or cell death[2]. Non-homologous end-joining (NHEJ) is required to attach chromosomal DSB ends together in mammals[3]. While detailed structural and mechanistic understanding of the NHEJ pathway is limited, error-prone X-family polymerases (pols) λ, μ, and Terminal Deoxynucleotidyl transferase (Tdt) are known to perform break synthesis during DSB repair[4]. Unlike most other polymerases, pol μ lacks nucleotide sugar selectivity and inserts both deoxy- (dNTPs) and ribonucleotides (rNTPs) into gapped DNA at similar catalytic efficiencies in vitro[5–9]. Recent work suggests pol μ-mediated rNTP insertion during repair synthesis is a requirement for efficient NHEJ in vivo[10]. High cellular rNTP/dNTP ratios (~100:1) promote ribonucleotide insertion[11,12], where stringent polymerase discrimination averts mutagenesis, DSB formation and other damage induced by genomic ribonucleotides[13].

Oxidation of cellular rNTP pools generates oxidized ribonucleotides, such as 7,8-dihydro-8-oxo-guanosine (8-oxo-rGTP)[14,15], that can be inserted into the genome by polymerases[16–21] (Fig. 1a). While limited information is available on the role and impact of damaged ribonucleotides on DNA replication and repair, insertion of oxidized dGTP (8-oxo-dGTP) by polymerases is known to be mutagenic[22,23]. We previously found that 8-oxo-dGTP can base pair with template adenine ($A_t$) in the pol μ active site upon rotation about the glycosidic bond into the mutagenic syn-conformation[24]. Insertion opposite template cytosine ($C_t$) in anti-conformation was enabled by active site binding of a third or product metal[25–29], that decreased 8-oxo-dGTP discrimination, such that pol μ efficiently inserted 8-oxo-dGTP opposite either template base. Elevated genomic 8-oxo-dG levels have been implicated in development of cancer, aging and human disease[30], as well as in bacterial antibiotic resistance[31–33]. Elaborate defense mechanisms have therefore evolved to suppress 8-oxo-dG accumulation in the genome. Oxidized dNTP pools are actively sanitized by enzymes such as MTH1 pyrophosphorylase, that degrades 8-oxo-dGTP, but not 8-oxo-rGTP, into the monophosphate, thereby limiting incorporation[34]. Oxidized ribonucleotide pool sanitization in mammals is poorly understood, and either does not occur, or the associated pathways have yet to be uncovered. Repair pathways that act on 8-oxo-dG, such as base-excision repair, inefficiently remove 8-oxo-rG containing lesions (Fig. 1b). Additionally, ribonucleotide excision pathways, such as ribonuclease H2A (RNaseH2A)-mediated excision repair[35], as well as base and nucleotide[36] excision repair, or Top1-mediated ribonucleotide repair[37], process 8-oxo-rG containing

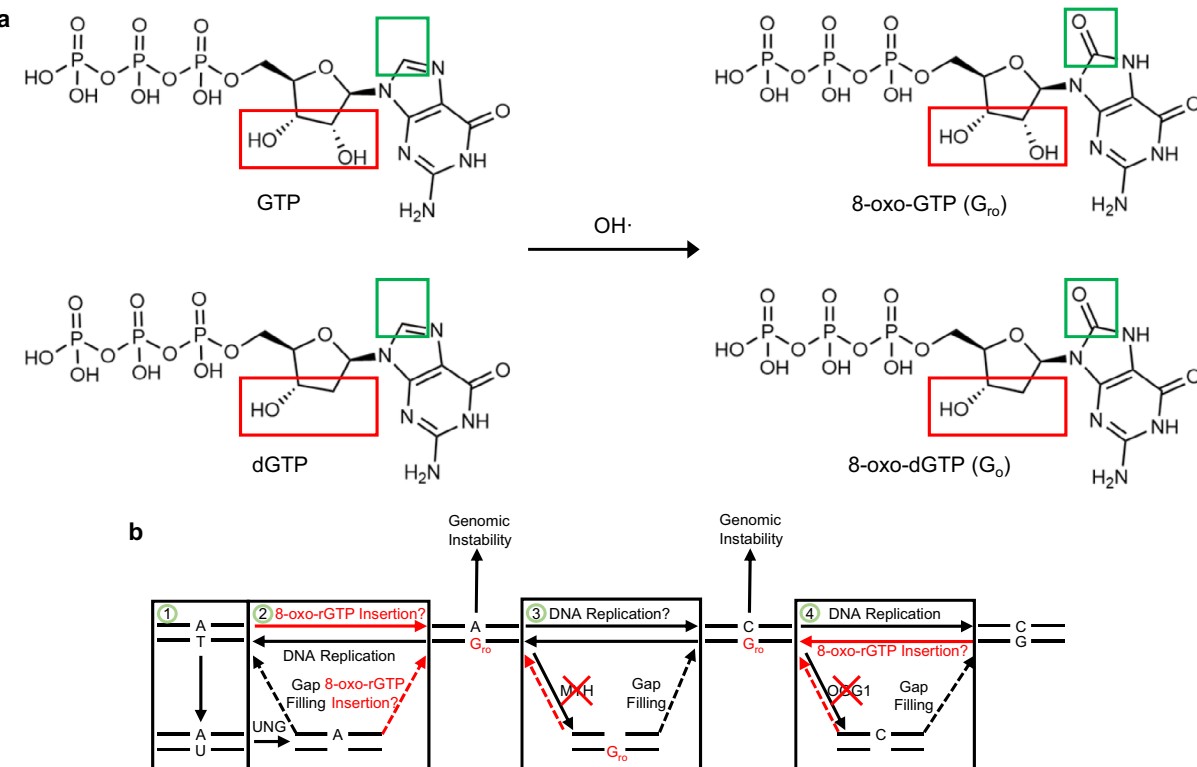

**Fig. 1 Repair pathways targeting deoxy-8-oxo-G inefficiently process the structurally equivalent ribonucleotide, 8-oxo-rG. a** Hydroxyl radicals (OH•) interact with and damage cellular free dGTP (2′-deoxyguanosine-5′-triphosphate) and rGTP (guanosine-5′-triphosphate) pools. Oxidation by hydroxyl radicals at C8 (green box) of dGTP and rGTP (red box) generates 8-oxo-dGTP (8-oxo-2′-deoxyguanosine-5′-triphosphate, 8dOG or $G_o$) and 8-oxo-rGTP (8-oxo-guanosine-5′-triphosphate, 8rOG or $G_{ro}$), respectively. **b** Scheme depicting known pathways associated with replication and repair of deoxy-8-oxo-G lesions that operate inefficiently on ribo-8-oxo-G lesions. (1) Thymine demethylation or UTP insertion, among other processes, may generate a genomic A-U base pair. The uracil can be removed by UNG (uracil DNA glycosylase, UDG) to generate a gapped repair intermediate. (2) The resulting gapped intermediate may undergo gap-filling by DNA polymerases and insertion of TTP or 8-oxo-rGTP opposite A to yield the Watson-crick (A-T) or A-$G_{ro}$ base pairs. Pols δ and ε are effectively unable to insert 8-oxo-rGTP and would likely not generate cytotoxic A-$G_{ro}$ intermediates. (3) MutYH-dependent repair of $G_o$ lesions can efficiently remove A opposite $G_o$, however, this pathway inefficiently excises A opposite $G_{ro}$ and therefore likely does not participate in repair of A-$G_{ro}$ base pairs. (4) OGG1 can remove $G_o$ opposite C from C-$G_o$ lesions, but is inefficient in removing $G_{ro}$ opposite C, so this pathway is unlikely to operate in repair of C-$G_{ro}$ lesions. These pathways promote buildup of $G_{ro}$ containing lesions that are potentially more cytotoxic than $G_o$ lesions and may lead to unrepaired genomic instability.

lesions inefficiently[38]. The large size of the readily oxidizable cellular rGTP pool, combined with lack of sanitization or repair pathways, implicates 8-oxo-rGTP insertion with high mutagenic potential in the active site of a polymerase that can proficiently incorporate rNTPs during its biological functions. Yet, 8-oxo-rGTP insertion by pol µ in DSB repair has not been previously examined.

Here we show that pol µ incorporates 8-oxo-rGTP into a model DSB substrate opposite $A_t$ with similar efficiency to 8-oxo-dGTP, but discriminates against insertion opposite $C_t$. Time-lapse crystallography snapshots of the insertion reactions in crystallo revealed that active site binding of 8-oxo-rGTP opposite $C_t$ occurred in a noncatalytic orientation. In contrast, binding opposite $A_t$ was accommodated in the mutagenic syn-conformation resulting in efficient insertion. Intermediates along the nucleotide insertion pathway demonstrated differences in the deoxyribose- and ribose nucleotide sugar conformations (Fig. 1a), as well as in active site substrate, metal, and side chain dynamics, that influence discrimination of 8-oxo-rGTP insertion. Pol µ-mediated 8-oxo-rGTP insertion during DSB repair is implicated as a source of mutagenesis and genomic instability.

## Results

**Active site instability of 8-oxo-rGTP opposite cytosine.** Most DNA polymerases (pols) discriminate against ribonucleotide (rNTP) insertion in order to prevent the mutagenesis and genomic instability associated with increased genomic ribonucleotides. X-family pol µ[8], however, has been reported to lack nucleotide sugar discrimination in the presence of $Mg^{2+}$. Kinetic analysis indicated that dGTP and rGTP insertion efficiencies were roughly equivalent opposite templates C ($C_t$) and A ($A_t$) (Fig. 2a, Supplementary Table 1, 2)[6]. Since pol µ has been suggested to employ $Mn^{2+}$ as the physiological metal[39,40], we also characterized insertion with $Mn^{2+}$ (Fig. 2a, Supplementary Table 2). Efficiency of undamaged nucleotide insertion was strongly increased opposite both templates, but discrimination remained unaltered. We previously showed[24] that $Mn^{2+}$ increases the efficiency of oxidized dGTP (8-oxo-dGTP) insertion opposite $C_t$, but modestly opposite $A_t$, compared to $Mg^{2+}$. Oxidized rGTP (8-oxo-rGTP) was preferentially inserted opposite $A_t$ in a single-nucleotide gap, and poorly opposite $C_t$ with $Mn^{2+}$. Efficiency opposite $C_t$ decreased by almost four orders of magnitude relative to rGTP insertion in the presence of $Mg^{2+}$. Replacing $Mg^{2+}$ with $Mn^{2+}$ rescued this decrease for 8-oxo-dGTP insertion[24], but only partially for the ribonucleotide. The presence of the 2′OH group, in combination with oxidized base damage (Fig. 1a, red box), therefore induces a switch in pol µ discrimination against insertion opposite $C_t$.

To provide a structural basis for these observations, we turned to time-lapse X-ray crystallography. We grew pol µ-DNA binary complex crystals bound to a DSB repair intermediate (Supplementary Fig. 1) with a templating C ($C_t$)[9]. We soaked these binary complex crystals in a cryo-solution containing 20 mM $Ca^{2+}$ and 2 mM 8-oxo-rGTP for 120 min to generate the $Ca^{2+}$-ground state (GS) 8-oxo-rGTP:$C_t$ ternary complex (Fig. 2b, Supplementary Table 3). The active site was nearly identical to the corresponding 8-oxo-dGTP(anti):$C_t$ ground state ternary complex (Fig. 2c). Key differences were observed, however, in DNA and nucleotide substrates (Fig. 2b, c). As we previously showed, 8-oxo-dGTP forms hydrogen bonds with $C_t$ in the anti-conformation, where O3′ is poised to attack $P_\alpha$ in the precatalytic ternary complex[24]. While the triphosphate of the incoming 8-oxo-rGTP was bound to the active site, stabilized by interactions with Gly320, Arg323, Lys325, and His329, density for the sugar and base was absent (Fig. 2b). O3′, along with the primer

terminal nucleotide, had rotated into the major groove, and a water molecule completed the coordination of $Ca^{2+}$ in the catalytic metal site ($Ca_c$) (Fig. 2b). The primer terminus rotation induced a ~2.5 Å shift in the palm subdomain and loop 2, as well as the primer and downstream template strands (Supplementary Fig. 2a, b).

8-oxo-rGTP binding to the polymerase active site is increased by $Mn^{2+}$ (Fig. 2a, Supplementary Tables 1, 2). We therefore soaked $Ca^{2+}$-bound precatalytic ternary complex crystals in a cryo-solution to exchange $Ca_c$ and $Ca_n$ for $Mn^{2+}$. This initiated catalysis of nucleotide insertion in crystallo. We froze the crystal after 15 min of soak and solved the structure of the resulting $Mn^{2+}$-ground state (GS) 8-oxo-rGTP:$C_t$ ternary complex (Fig. 2d, Supplementary Table 3). While the catalytic ($Ca_c$) and nucleotide ($Ca_n$) metal sites had undergone exchange for $Mn^{2+}$ (Supplementary Fig. 2c), density for a bond between O3′ and $P_\alpha$ was not observed (Fig. 2d). Density for the 8-oxo-rG base was observed, but in an inverted orientation from that in the $Ca^{2+}$-ground state 8-oxo-dGTP(anti):$C_t$ ternary complex (Supplementary Fig. 2d). The 8-oxo-rG base pointed outward into the major groove and was stabilized by His329 (Fig. 2d). The primer terminus ($P_n$) still occupied the rotated conformation. Lys438 had altered conformation to interact with 8-oxo-rGTP (Fig. 2d, Supplementary Fig. 2d). Global conformational changes compared to the deoxy structure were identical to the $Ca^{2+}$-GS complex (Supplementary Fig. 2a, b). Density for Lys438 appeared in multiple locations and was too weak to model in the $Ca^{2+}$-GS complex (Fig. 2b).

**Snapshots of insertion opposite cytosine.** The effect of 2′OH on $k_{cat}$ suggested differences between 8-oxo-rGTP and 8-oxo-dGTP insertion. To investigate this possibility, we obtained snapshots of the insertion reaction in crystallo (Fig. 3a, b, c, Supplementary Table 3). After 960 min of soak in a cryo-solution containing 50 mM $Mn^{2+}$, ~50% of incoming 8-oxo-rGTP had been incorporated opposite $C_t$ in anti-conformation in the $Mn^{2+}$-reaction state (RS) 8-oxo-rGTP(anti):$C_t$ ternary complex (Fig. 3a). Simulated annealing omit ($F_o$-$F_c$) density indicated phosphate inversion and bond formation had occurred, and alternate conformations of the incoming nucleotide and primer terminus were present at ~50% occupancy (Fig. 3a, Supplementary Fig. 3a, b). 8-oxo-rGMP(anti) in the reacted conformation forms short (2.4–2.8 Å) hydrogen bonds with $C_t$. O2′ clashes (2.3 Å) with a backbone carbonyl oxygen of Gly433 and O8 with an oxygen of $P_\alpha$ (Fig. 3b). Interactions with Lys325 and Arg445 stabilize 8-oxo-rGMP(anti) similarly to the deoxynucleotide. $PP_i$ was assumed to be present (Fig. 3c) and was modeled into the density in a conformation expected directly after cleavage. The catalytic ($Mn_c$) and nucleotide ($Mn_n$) metal sites contained $Mn^{2+}$ (Supplementary Fig. 3c). Anomalous density for additional metals was absent, however, simulated annealing ($F_o$-$F_c$) omit density indicated the presence of an atom in the expected location of the previously observed third or product metal ($Mn_{p,8dOG}$; Fig. 3a, Supplementary Fig. 3c). The low occupancy (~25%) atom coordinated O8, an oxygen of the incorporated phosphate, and an oxygen of $PP_i$ with long (~2.4–2.5 Å) coordination distances. The coordination distances and geometry suggest the site may be occupied by $Na^+$, or a low occupancy $Mn^{2+}$ ($Mn_{p,8rOG}$). The position of $Mn_{p,8rOG}$ differed from that of $Mn_{p,8dOG}$ observed in the 8-oxo-dGTP(anti):$C_t$ insertion, however, and was located ~0.7 Å into the major groove (Fig. 3b (inset), Supplementary Fig. 3c). This difference likely resulted from an altered ribose sugar conformation, influencing positioning of O8 (Fig. 3b). The $Mn^{2+}$-reaction state (RS) intermediate was otherwise identical to the $Ca^{2+}$- and $Mn^{2+}$-GS complexes.

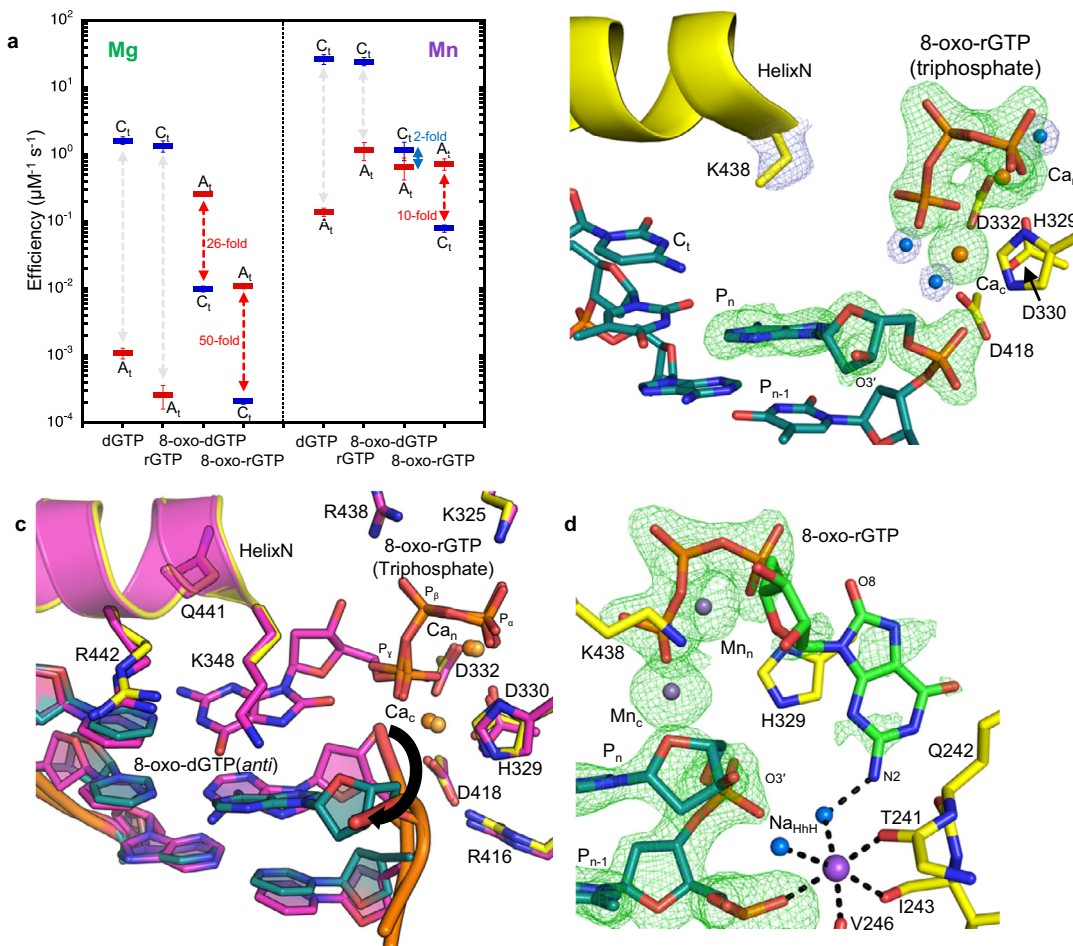

**Fig. 2 Structural basis for 8-oxo-rGTP discrimination opposite cytosine. a** Kinetic analysis of oxidized and undamaged (d)GTP insertion by pol μ. Catalytic efficiency is indicated as a blue line for insertion opposite template base C ($C_t$) and as a red line for insertion opposite template base A ($A_t$), in the presence of $Mg^{2+}$ (left panel) or $Mn^{2+}$ (right panel). Fold preference is indicated next to a blue ($C_t$) or red ($A_t$) dashed arrow. The error bars represent standard errors (S.E.) derived from three independent measurements. **b** Active site of the $Ca^{2+}$-bound precatalytic ground state 8-oxo-rGTP:$C_t$ ternary complex. Protein sidechains are shown in yellow stick representation and DNA is in cyan, Helix N is shown as a yellow cartoon. $Ca^{2+}$ atoms are the orange spheres, waters are blue spheres. Simulated annealing ($F_o$–$F_c$) omit density (green mesh) shown is contoured at 3 σ. The $\sigma_A$ (2$F_o$–$F_c$) map shown as a blue mesh is contoured at 1.5 σ with a carve radius of 1.0 Å. **c** Comparison of $Ca^{2+}$-bound precatalytic ground state deoxy- (purple sidechains and DNA) and ribo-8-oxo-GTP:$C_t$ (yellow sidechains and cyan DNA) ternary complexes. Rotation of the primer terminus ($P_n$) in the 8-oxo-rGTP structure is shown with a blackarrow, placing O3´ in an inverted orientation that is incompatible with attack at $P_\alpha$. **d** $Mn^{2+}$-ground state 8-oxo-rGTP:$C_t$ ternary complex (15 min soak) showing 8-oxo-rGTP in an unreactive orientation. $Mn^{2+}$ atoms are the small purple spheres, $Na^+$ is shown as a larger purple sphere.

Near complete 8-oxo-rGTP(anti):$C_t$ insertion (>80% conversion) had occurred after 2160 min of soak (Fig. 3c, Supplementary Table 3). Density near the third metal location was no longer observed in this $Mn^{2+}$-product state (PS) ternary complex. P1 of $PP_i$ (former $P_\beta$ of 8-oxo-rGTP) was still bound, while reduced simulated annealing omit ($F_o$–$F_c$) density for P2 of $PP_i$ (former $P_\gamma$ of 8-oxo-rGTP) was present, precluding accurate modeling. The active site remained otherwise identical to the $Mn^{2+}$-reaction state and $Mn_n$ and $Mn_c$ were still bound (Supplementary Fig. 4a). Overlay of the ribo- and deoxy-8-oxo-dGTP(anti):$C_t$ product (PS) complexes indicated they were largely identical, apart from the absence of $Mn_{p,8rOG}$ (Supplementary Fig. 4a, b). $Mg_c$ and $Na_n$ were bound after a 2160 min soak in a cryo-solution containing 50 mM $Mg^{2+}$ and density for incoming or incorporated nucleotide was absent (Supplementary Fig. 4c, Supplementary Table 3).

To gain insight into active site features that stabilize the unreactive and reacted conformations, interaction free energies were calculated using molecular dynamics simulations (Fig. 3d, Supplementary Tables 6, 7). The total free energy for the unreactive

orientation of 8-oxo-rGTP (−178 ± 12 kcal mol⁻¹) was similar to the reacted conformation (−172 ± 10 kcal mol⁻¹). Analysis of residue contributions indicated that the majority of stabilization of each conformation was provided by active site metals and residues that interact directly with the triphosphates or nearby residues (Thr318–His329). The major differences between conformations was provided by Lys325 for the unreactive orientation (−34.5 kcal mol⁻¹) compared to the reacted active site (−5.2 kcal mol⁻¹). Sugar stabilization was provided by Trp434 in the reacted active site (−4.4 kcal mol⁻¹ compared to 0.8 kcal mol⁻¹), while Lys438 stabilized the ribose sugar in the unreactive orientation (−17.3 kcal mol⁻¹ compared to −6.7 kcal mol⁻¹), consistent with that observed in the $Mn^{2+}$-ground state 8-oxo-dGTP(anti):$C_t$ ternary complex[24]. While Gln441 does not directly interact with either incoming oxidized deoxy- or ribonucleotide (Fig. 2c), increased stabilization of the reacted conformation was observed (−9.8 kcal mol⁻¹ compared to −0.5 kcal mol⁻¹). The N-terminal domain provided additional stabilization (~5–12 kcal mol⁻¹) of the unreactive conformation, while interactions with DNA were

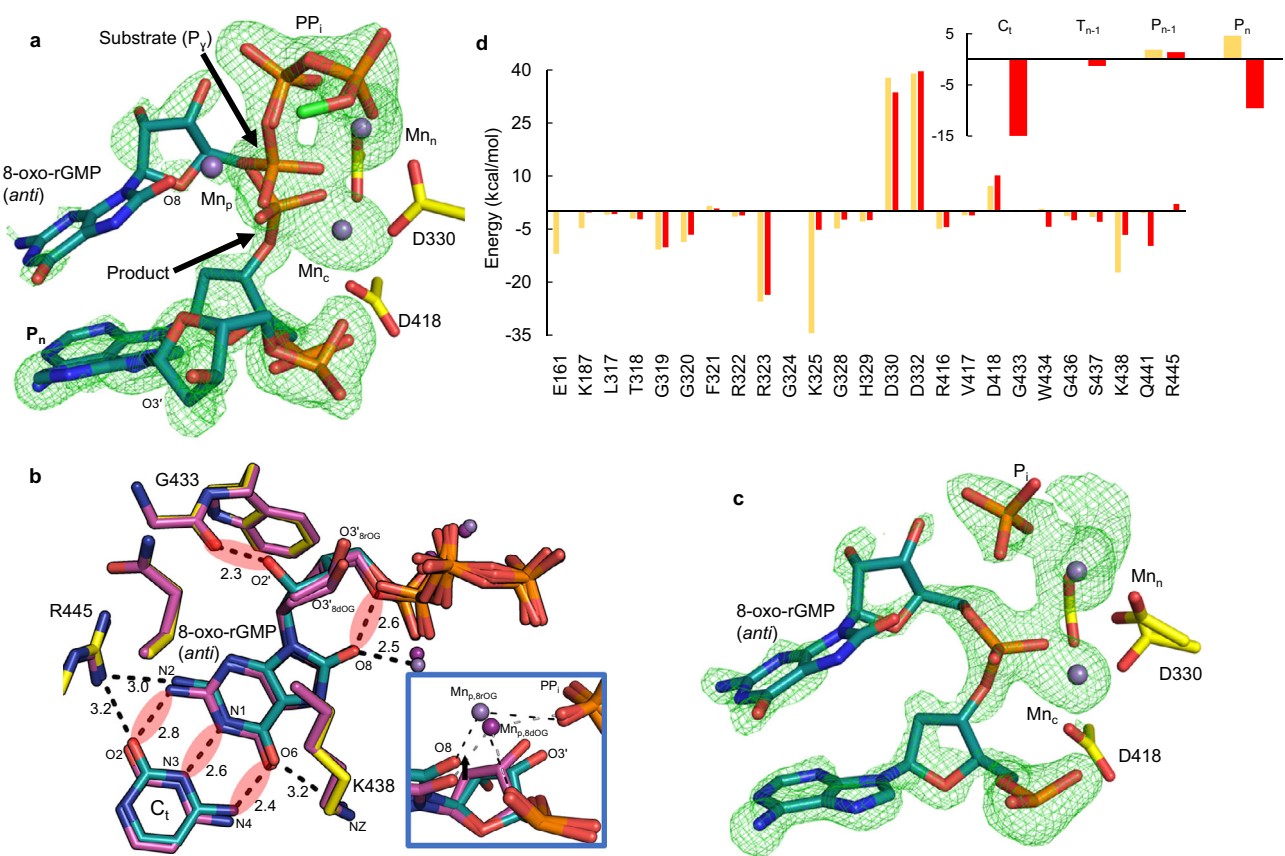

**Fig. 3 8-oxo-rGTP(anti) insertion opposite cytosine. a** Active site of the reaction state ternary complex of the 8-oxo-rGTP(anti):$C_t$ insertion after an overnight soak (960 min) in a $Mn^{2+}$-containing cryo-solution. Arrows indicate bond broken (substrate) and formed (product). Protein sidechains are shown in yellow stick representation, nucleotide in green and DNA in cyan. $Mn^{2+}$ is shown as a magenta sphere. Simulated annealing ($F_o$–$F_c$) omit density (green mesh) is shown at 3 σ. **b** Top down close up view of nucleotide and possible product metal ($Mn_{p,8rOG}$, purple sphere) interactions in the ribo-8-oxo-GMP(anti)/GTP:$C_t$ active site. Shown is an overlay with the deoxy-8-oxo-GTP(anti)/GMP(anti):$C_t$ $Mn^{2+}$-reaction state ternary complex in magenta. The product metal in the latter structure ($Mn_{p,8dOG}$) is shown as a magenta sphere. $Mn_{p,8rOG}$ bridges O8, an oxygen of the phosphate of the incorporated nucleotide and an oxygen of $PP_i$. Hydrogen bonding and other key distances (Å) are shown with black dashes. Red spheres indicate (unfavorable) destabilizing interactions. Inset. Comparison of $Mn_{p,8rOG}$ and $Mn_{p,8dOG}$ coordination of O8 and product phosphate oxygens. A black arrow indicates differences in the position of O8. **c** $Mn^{2+}$-product complex of the 8-oxo-rGTP(anti):$C_t$ insertion. Simulated annealing omit density is shown as a green mesh contoured at 3 σ. **d** Contributions of selected protein and nucleic acid residues to the interaction free energy of 8-oxo-rGTP in the unreactive (light orange) and reacted (red) conformations within the $Mn^{2+}$-reaction state ternary complex of the 8-oxo-rGTP:$C_t$ insertion (see Supplementary Tables 6, 7). $C_t$ (template nucleotide), $T_{n-1}$ (opposite $P_{n-1}$ in template strand), $P_{n-1}$ (primer strand), and $P_n$ (primer terminal nucleotide) represent selected nucleotides in the template or primer strands.

destabilizing ($P_n$, 4.6 kcal mol$^{-1}$; $P_{n-1}$, 1.9 kcal mol$^{-1}$; Fig. 3d (inset), Supplementary Table 7). The N-terminal domain had less of a stabilizing effect on the reacted conformation (~0 kcal mol$^{-1}$), but DNA appeared to have a more stabilizing effect ($T_n$, −15 kcal mol$^{-1}$; $P_n$, −9.6 kcal mol$^{-1}$). Overall, these results suggest that either conformation may be adopted in the ground state ternary complex, consistent with structural observations.

**Charge dynamics stabilizes oxidized nucleotides opposite adenine.** The active site of the $Ca^{2+}$-bound 8-oxo-rGTP:$A_t$ pre-catalytic ground state (GS) ternary complex was largely identical to that observed previously for 8-oxo-dGTP(syn):$A_t$ (Fig. 4a, b, Supplementary Table 4)[24]. The 8-oxo-rG base forms hydrogen bonds with $A_t$ in the syn-conformation using its Hoogsteen edge. Lys438 forms van der Waals interactions with the base and provides a potential hydrogen bonding partner for O6. N2 and O1 of $P_\alpha$ form a stabilizing hydrogen bond to securely anchor the base in the nucleotide-binding pocket. Arg445 stabilizes the template base, and the latter superimposes with the template base in the $Ca^{2+}$-ground state 8-oxo-dGTP(syn):$A_t$ ternary complex (Fig. 4b). The conformations of the sugar, base, and triphosphate

of 8-oxo-rGTP(syn) appear to differ substantially from 8-oxo-dGTP(syn). The rigid ribose sugar pucker (C4′ exo) forces the 8-oxo-rG(syn) base to shift relative to 8-oxo-dG(syn), such that O8 is displaced by ~0.6 Å. This shift appears to result from O2′ interaction with the backbone carbonyl oxygen of Gly433 yielding a shifted Trp434, that now accommodates the rigid ribose sugar. Apart from the shifted base and altered sugar conformation, the electrostatic environment of $P_\beta$, and thus triphosphate stabilization, is substantially impacted such that an oxygen of $P_\beta$ (O2B) that interacts with O3′ in the deoxy structure is now ~0.7 Å closer.

We also determined the structure of the $Mn^{2+}$-ground state (GS) 8-oxo-rGTP(syn):$A_t$ ternary complex 15 min after initiating the reaction by soaking $Ca^{2+}$-GS crystals in a cryo-solution containing 50 mM $Mn^{2+}$ (Supplementary Fig. 5a, b, Supplementary Table 4). The resulting $Mn^{2+}$-ground state ternary complex was similar to the $Ca^{2+}$ complex, except for exchange of $Ca_n$ and $Ca_c$ for $Mn^{2+}$, and rotation of the primer terminus to the position observed opposite $C_t$. A 15-min soak performed with 8-oxo-dGTP(syn):$A_t$ $Ca^{2+}$-GS crystals lacked bond formation, primer terminus rotation and displayed an additional $Mn^{2+}$ ($Mn_D$)

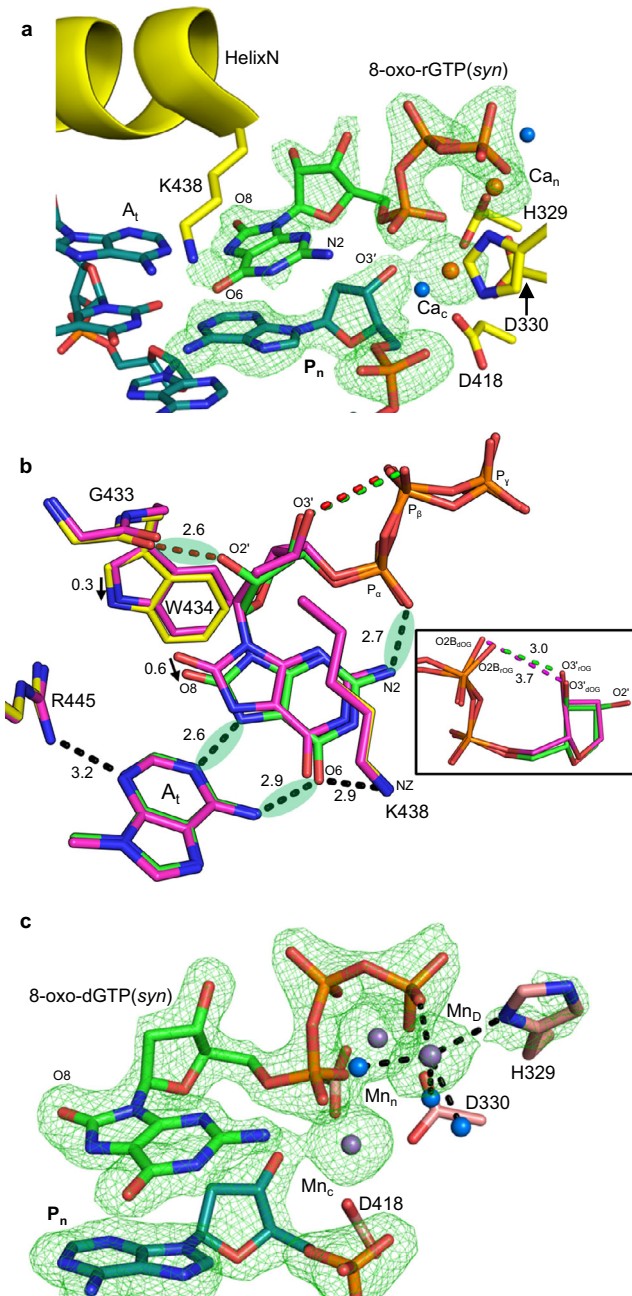

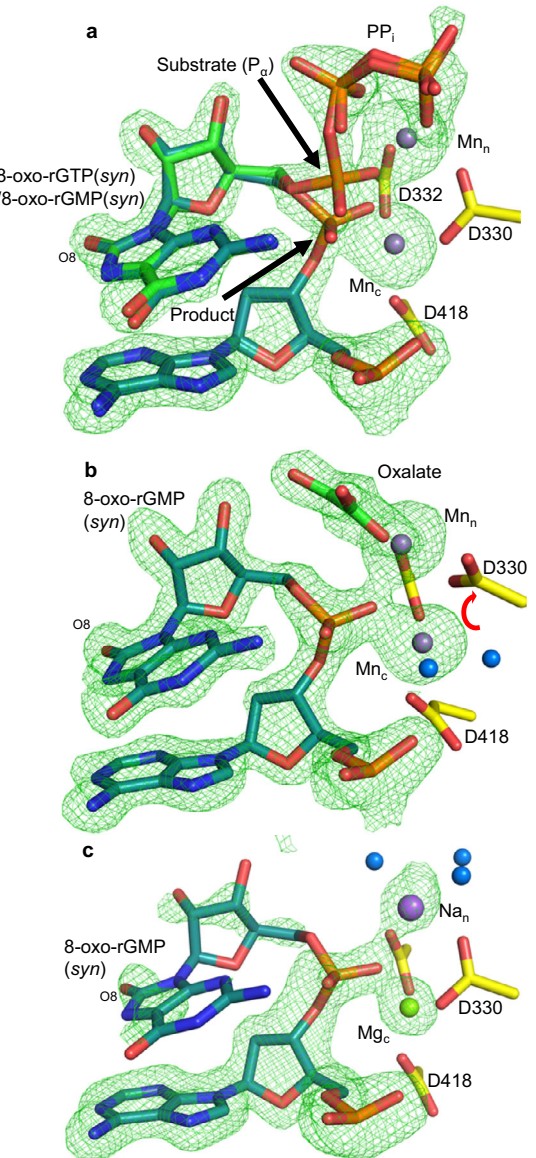

**Fig. 4 Ground state 8-oxo-rGTP(syn):$A_t$ ternary complex. a** Active site of the $Ca^{2+}$-bound pre-catalytic ground state 8-oxo-dGTP(syn):$A_t$ ternary complex. Protein sidechains (yellow), DNA (cyan), and nucleotide (green) are shown in stick representation, Helix N is the yellow cartoon. $Ca^{2+}$ atoms are the orange spheres, water molecules are blue spheres. Simulated annealing omit density (green mesh) is contoured at 3 σ. **b** Comparison of active site nucleotide interactions in the $Ca^{2+}$-ground state 8-oxo-rGTP (syn):$A_t$ (green and yellow) and 8-oxo-dGTP(syn):$A_t$ (magenta) ternary complexes. Hydrogen bonding (Å) and key distances (Å) are shown with black and red dashes, respectively. Green spheres indicate (favorable) stabilizing interactions. The inset shows that the distance between O3′ and O2B is altered due to the rigid sugar pucker induced by the ribose (O2′) oxygen. **c** $Mn^{2+}$-ground state 8-oxo-dGTP(syn):$A_t$ ternary complex (15 min soak). An additional metal ($Mn_D$) coordinates Pγ, His329 and 3 water molecules. Black dashes show metal coordination.

**Fig. 5 8-oxo-rGTP(syn) insertion opposite adenine. a** Active site of the $Mn^{2+}$-reaction state (RS) ternary complex of pol μ in the process of inserting 8-oxo-rGTP(syn) opposite $A_t$. Arrows indicate bond broken (substrate) and formed (product). Protein sidechains are shown in yellow stick representation, the incoming nucleotide is green and DNA is cyan. $Mn^{2+}$ atoms are shown as magenta spheres. Simulated annealing omit density is shown as a green mesh contoured at 3 σ. **b** Active site of the 8-oxo-rGMP(syn):$A_t$ $Mn^{2+}$-product (PS) complex. A red arrow indicates rotation of Asp330 compared to the ground and reaction states. Water molecules are shown as blue spheres. **c** Post-catalytic (960 min) $Mg^{2+}$:8-oxo-rGMP(syn):$A_t$ product (ES) complex. The larger purple sphere is $Na^+$, while $Mg^{2+}$ is a green sphere.

coordinated by Pγ, an altered His329 and three water molecules (Fig. 4c, Supplementary Fig. 5c, Supplementary Table 4).

**Snapshots of insertion opposite adenine.** A differential effect of the 2′OH group on the efficiency of 8-oxo-rGTP:$A_t$ and 8-oxo-dGTP:$A_t$ insertions (<10-fold) was not apparent based on kinetic analysis (Fig. 2a, Supplementary Tables 1, 2). While 8-oxo-rGTP

(syn) is preferentially inserted opposite $A_t$ in a single-nucleotide gap, that is, almost as efficiently as 8-oxo-dGTP(syn) in the presence of $Mn^{2+}$, efficiency decreased by roughly two orders of magnitude compared to 8-oxo-dGTP(syn) in the presence of $Mg^{2+}$ (Fig. 2a). Occupancy refinement indicated that after 30 min of soak in cryo-solutions containing either 50 mM $Mn^{2+}$ (Fig. 5a, Supplementary Table 4) or $Mg^{2+}$ (Supplementary Fig. 6a, Supplementary Table 5), ~50% of incoming 8-oxo-rGTP(syn) had been incorporated opposite $A_t$. The catalytic and nucleotide metal sites were occupied by $Mn^{2+}$ in the $Mn^{2+}$ soak (Supplementary Fig. 6b), and $Mg^{2+}$ in the $Mg^{2+}$ soak (Supplementary Fig. 6a), while the position of $PP_i$ was as expected directly after bond cleavage (Fig. 5a, Supplementary Fig. 6a). Full 8-oxo-rGTP(syn): $A_t$ insertion was observed after 120 min of soak in either $Mn^{2+}$ (Fig. 5b, Supplementary Table 4) or $Mg^{2+}$ (Supplementary Fig. 6c, Supplementary Table 5) containing cryo-solutions. $Mn_n$ and $Mn_c$ were still present, but an additional water molecule coordinated $Mn_c$, as Asp330 had rotated ~90° (Fig. 5b). Other active site features remained identical to the $Mn^{2+}$-reaction state complex. $Mg_n$ and $Mg_c$ were still bound in the $Mg^{2+}$ product complex, but Asp330 remained in the ground state conformation. Post-catalytic (~960 min) soaks of the product complexes indicated the newly formed base pair remained only partially intact (Fig. 5c, Supplementary Fig. 6d, Supplementary Tables 4, 5). Density for the 8-oxo-rG(syn) base was reduced in the $Mg^{2+}$ soak, while $A_t$ had shifted ~1.2 Å and displayed an altered conformation compared to the $Mg^{2+}$-PS complex (Supplementary Fig. 6e). $PP_i$ had dissociated, and water molecules coordinated $Na_n$ that was bound at reduced (~50%) occupancy (Supplementary Table 5, Fig. 5c). $Mg_c$ was still bound, but at reduced (~60%) occupancy. $Mn_n$ and $Mn_c$ were still bound in the $Mn^{2+}$ soaks and $A_t$ had not shifted (Supplementary Fig. 6d). Similar soaks of the 8-oxo-dGMP product complex displayed an intact base pair[24]. These observations indicate increased dynamics of the nascent 8-oxo-rG(syn):$A_t$ primer terminus.

## Discussion

In this study, time-lapse crystallography uncovers a role for 8-oxo-rGTP insertion during pol μ-mediated double strand break synthesis, circumventing defenses against both ribonucleotide and oxidized nucleotide insertion. Lacking known repair or breakdown pathways (Fig. 1b), oxidized ribonucleotide insertion represents an emerging and persistent threat to genomic stability.

### Ribonucleotide discrimination

Genomic ribonucleotide incorporation can lead to replication blockage, mutagenesis and DSB formation. Due to the high cellular rNTP/dNTP imbalance, discrimination against ribonucleotide insertion by most DNA polymerases is essential for the maintenance of genomic integrity. Discrimination against substrates bearing 2′OH groups (Fig. 1a) is attributed to two aromatic residues (Tyr and Phe) lining the minor groove nucleotide-binding pocket in X-family pols β[41] and λ[42]. Functioning as a pair, these residues interfere with active site stability of the incoming ribonucleotide or mismatch. Decreased discrimination (Fig. 2a)[5,6,10] has been attributed to a modified steric gate in pol μ (Gly and Trp). Equivalent deoxyribose and ribose stabilization by Trp434 eliminates stringent sugar selection (Figs. 2a, 3b, 4b). O2′ severely (~2.3 Å, Fig. 3b) clashes with the backbone carbonyl oxygen of Gly433 in the anti-, but not syn-conformation (~2.6 Å, Fig. 4b)[6]. Lack of an aromatic residue replacing Gly433 likely promotes circumvention of matched base-pair checking, promoting increased mismatch incorporation. Rigid positioning of the template strand and incoming triphosphate requires rotation of $A_t$ into syn-conformation during

dGTP(anti):$A_t$(syn) misinsertion[24], and likely during rGTP:$A_t$ misinsertion (Fig. 2a), presenting kinetic and energetic barriers to catalysis. The disruption of template strand base stacking interactions may induce strain that results in reduced efficiency of rGTP:$A_t$ insertion.

### Discrimination against 8-oxo-rGTP insertion opposite cytosine

Oxidized deoxynucleotides, such as 8-oxo-dGTP, are mutagenic due to their ability to hijack fidelity determinants employed by DNA polymerases to discriminate against insertion of noncanonical nucleotides. Insertion opposite $C_t$ is thus much less efficient compared to insertion opposite $A_t$ for many DNA polymerases[43]. Although 8-oxo-dGTP can be accommodated in the pol μ active site in the syn-conformation opposite $C_t$, rotation about the glycosidic bond into anti-conformation is required for insertion[24]. Similarly, 8-oxo-rGTP:$C_t$ insertion is unfavorable, in part, due to the requirement to adopt the anti-conformation (Fig. 3). The stability of the syn-conformation arises from positioning of the C8 oxygen away from ribose or deoxyribose oxygens (Fig. 4b), whereas the anti-conformation forces these oxygens into close proximity (Fig. 3b). Pol β requires a ground state metal to neutralize the resulting clash and stabilize the anti-conformation in the $Ca^{2+}$-(PDB id 4UB4) and $Mn^{2+}$-(PDB id 4UB5) ground state 8-oxo-dGTP(anti):$C_t$ ternary complexes[44]. Due to its rigid active site and an altered nucleotide conformation, pol μ has no need for such a metal[24].

These observations appear not to extend to 8-oxo-rGTP(anti) insertion opposite $C_t$, however, as 8-oxo-rGTP binding occurs in an unreactive orientation (Fig. 2b, d). Capture of O3′ and nucleophilic attack at $P_α$ forces 8-oxo-rGTP to adopt the reacted conformation and base pair with $C_t$ in the anti-conformation (Fig. 3a). Both unreactive and reacted conformations are energetically equivalent and differentially stabilized by the active site (Fig. 3d, Supplementary Tables 6, 7). As expected, both conformations are strongly stabilized by the active site metals and triphosphate interacting sidechains. In particular, Lys325 interacts with the triphosphate of the unreactive conformation and Lys438 provides ribose stabilization (Fig. 2d, Supplementary Figs. 2d, 4b). While the different steric gate contributes to rNTP discrimination, additional factors appear to play roles in oxidized ribonucleotide discrimination by pol μ compared to other X-family pols. The rigid sugar pucker in combination with O8 creates a clash with the modified steric gate, and a very short O2′–Gly433 distance (~2.3 Å), promoting adoption of the unreactive conformation (Fig. 3b, d). Exceptionally short base-pairing distances in the canonical conformation (2.4 Å, N4–O6; 2.6 Å, N1–N3; 2.7 Å, N2–O2), in range for symmetric hydrogen bond lengths[45], in addition to other destabilizing interactions with DNA (Fig. 3d), additionally influence active site discrimination. These factors promote adoption of the unreactive conformation, that is also stabilized by other interactions, including with the DNA binding HhH motif (Fig. 2b, d). The rotated primer terminus significantly contributes to increased discrimination opposite $C_t$, as the primer terminus must reorient for O3′ to attack $P_α$(8-oxo-rGTP) (Supplementary Figs. 3a, b). This rotation constitutes a barrier for insertion and correlates with decreased insertion efficiency (Fig. 2a). The primer terminus displacement is thus modulated by active site features that provide stabilization of the unreactive and reacted conformations, and is reminiscent of features of the 8-oxo-dGTP(syn):$C_t$ ground state ternary complex of the Lys438Asp variant[24]. In the latter structure, however, 8-oxo-dGTP(syn) forms hydrogen bonds with $C_t$ in the reacted orientation without bond formation and is not bound in the unreactive orientation.

Primer terminus entry into the active site, perhaps due to thermal dynamics, promotes the nucleotidyl transferase reaction,

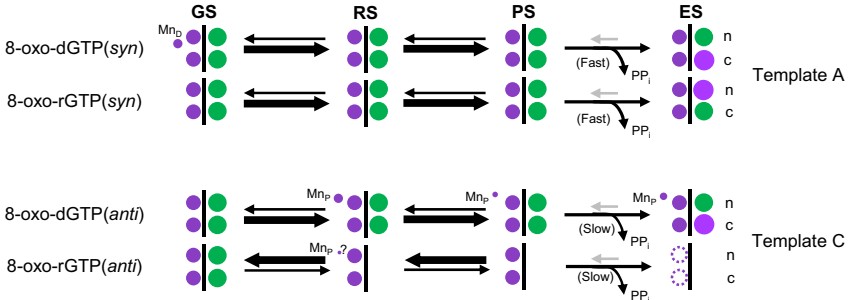

**Fig. 6 Active site metal modulation of ribo- and deoxy-8-oxo-GTP discrimination opposite adenine and cytosine.** Equilibrium between the ground (GS), reaction (RS), product state (PS) and extended soak (ES) intermediates during 8-oxo-dGTP and 8-oxo-rGTP insertion opposite $A_t$ and $C_t$ is shown. Thick and thin arrows indicate stronger and weaker preference for the given direction of reaction (forward or reverse), respectively. The smaller magenta and green circles depict the active site $Mn^{2+}$ and $Mg^{2+}$ atoms, respectively, while the larger purple spheres are $Na^+$ atoms. Dashed circles indicate intermediates yet to be observed. The letters to the right denote the catalytic (c) and nucleotide (n) metals. The third smaller magenta circle indicates observation of $Mn_p$ or $Mn_D$. $PP_i$ release is either "slow" or "fast" and is shown as a curved arrow, while a grey arrow indicates the possibility of $PP_i$ re-binding.

where $Mn_c$ (Fig. 3a) but not $Mg_c$ can capture O3′ (Supplementary Fig. 4c), resulting in strongly decreased insertion efficiency in the presence of $Mg^{2+}$ (Fig. 2a). The very short coordination distances of $Mn_c$ and Asp330 with the inserted phosphate ($P_{n+1}$) suggest decreased affinity of metals for the catalytic metal site that may be overcome by $Mn_c$, but not $Mg_c$, binding. Additionally, since evidence for $Mn_p$ in the reaction state included only a weak omit ($F_o–F_c$) peak, and anomalous density at this site was lacking, $Mn_p$ stabilization of the product complex is not as efficient as for 8-oxo-dGTP(anti):$C_t$. This likely promotes the reverse reaction and adoption of the unreactive conformations of the nucleotide and primer terminus (Fig. 6). Unreacted 8-oxo-rGTP eventually dissociates from the active site, as suggested by the lack of active site omit density in the 2160 min $Mg^{2+}$ soak (Supplementary Fig. 4c). These factors contribute toward insufficient stabilization of the product complex and inefficiency of this insertion in vitro (Fig. 2a). Similarly to the 8-oxo-dGTP(anti):$C_t$ insertion, the ribonucleotide influences active site $PP_i$ retention and dynamics. The absence of significant evidence for $Mn_p$ suggests delayed $PP_i$ release is perhaps an intrinsic feature of the anti-conformation, that is modulated by $Mn_p$ binding. $P_γ$ is released first after bond formation, as density for $P_β$ is still observed after 2160 min of soak (Fig. 3c).

**Hoogsteen base-pairing enables efficient 8-oxo-rGTP insertion opposite adenine.** The energetically favorable syn-conformation positions O8 away from ribose oxygens, enabling Hoogsteen base-pairing and preferential 8-oxo-rGTP(syn):$A_t$ insertion, similarly to 8-oxo-dGTP(syn) (Figs. 2a, 4a). The 2′OH group clashes less with the pol μ steric gate in the syn-conformation (~2.6 Å), promoting stability of the incoming ribonucleotide and primer terminus, leading to productive insertion (Fig. 4b). Differences in the deoxy- and ribonucleotide sugar conformations result in a shifted 8-oxo-rG(syn), while still maintaining hydrogen bonding distances (2.6 Å and 2.9 Å) equivalent to those observed for 8-oxo-dG(syn). A fourth metal in the $Mn^{2+}$-ground state 8-oxo-dGTP(syn):$A_t$ ternary complex, referred to here as metal D ($Mn_D$), stabilizes $P_γ$ in the absence of O2′ (Fig. 4c). This metal is absent in the $Mn^{2+}$:8-oxo-rGTP(syn):$A_t$ ground state structure, and the primer terminus can be modeled in both rotated and unrotated conformations (Supplementary Fig. 5a), suggesting $Mn_D$ binding influences primer terminus stability. This metal was not observed in the $Ca^{2+}$-ground state 8-oxo-dGTP:$A_t$ ternary complex of pol β (PDB id 4UAW)[44], but a product metal coordinated by $P_β$ stabilized the product state. Such a metal was absent in the product state structure for pol μ[24].

Curiously, $Mn_D$ coordinates $P_γ$ in a location close to that of a third metal observed in precatalytic ternary complexes of family B enzymes such as pol δ[46]. The 2′OH group thus induces differences in active site charge dynamics compared to 8-oxo-dGTP (syn), resulting in very similar insertion efficiencies. Higher energy intermediate states thus appear to precede bond formation by DNA polymerases. Additionally, the active site structure induced by 8-oxo-rG(syn) enables rapid $PP_i$ release compared to the anti-conformation (Fig. 5b, c) removing the requirement for product metal stabilization of the product complex. The effect of $Mn^{2+}$ is thus likely restricted to modulating occupancy of the catalytic metal site, where the catalytic metal clashes less with the phosphate of the nascent primer terminus (Supplementary Fig. 5b) than in the anti-conformation (Supplementary Fig. 4a). Since the $Mg^{2+}$ and $Mn^{2+}$ reaction and product state intermediates are very similar, the modified position of 8-oxo-rGTP (syn) compared to the deoxy insertion, combined with an effect on catalytic metal binding, likely decrease efficiency for this insertion (Fig. 2a). The lack of density for the base and sugar post-insertion suggests the 8-oxo-rG(syn):$A_t$ base pair dissociates or "frays". These potentially cytotoxic repair intermediates would require additional processing to either remove the lesion or ligate the dynamic ends.

While an intricate cellular defense network against oxidative DNA damage maintains genome stability and averts disease, these safeguards are ineffective against oxidized ribonucleotides. Considering the large cellular rNTP/dNTP imbalance, an enlarged substrate pool promotes generation of oxidative ribonucleotide damage through proficient pol μ-mediated 8-oxo-rGTP insertion[14]. RNaseH2A mediated post-synthetic excision repair (RER) would be expected to excise oxidized ribonucleotides, yet 8-oxo-rG excision is suppressed by base-excision repair[18,38]. MutY homologue (MYH) removes dA from the 8-oxo-dG:dA mispair, but does not remove rA[16]. Yeast Ogg2 can remove 8-oxo-dG opposite dA, but 8-oxo-rG removal opposite dA was not characterized[47]. 8-oxo-guanine DNA glycosylase (OGG1) also does not efficiently process 8-oxo-rG:C lesions[18]. Since 8-oxo-rG lesions cannot be removed without replicative dilution, the persistent nature of these cytotoxic lesions implicates pol μ mediated 8-oxo-rGTP insertion as a potential source of genomic instability[48].

**Metal dynamics in fidelity of oxidized ribonucleotide insertion.** Longer soaks of the 8-oxo-rGTP(anti):$C_t$ product complex indicate that $PP_i$ is retained in the active site post-insertion (Fig. 3c). The absence of $Mn_p$ (Fig. 3b), combined with the effect of the 2

´OH group on substrate dynamics, decreases the stability of the product complex during and after insertion. The rate of the reverse reaction is impacted, such that $Mn_p$ is not available to arrest the reverse reaction (Fig. 6). The vacant active site at the end of the long soak may thus indicate that the triphosphate eventually dissociates from the active site (Supplementary Fig. 4c). Alternatively, the product base pair may dissociate and become disordered, as observed for 8-oxo-rGTP(syn) insertion opposite $A_t$ (Fig. 5b, c). The probability of a product release event compared to the reaction proceeding in reverse thus influences fidelity and discrimination of 8-oxo-rGTP incorporation (Fig. 6). The probability of undergoing the reverse reaction is decreased for 8-oxo-dGTP(syn) and -rGTP(syn) insertion opposite $A_t$, where reduced density for $PP_i$ is observed even in the immediate product ternary complex (Fig. 5b, Supplementary Fig. 5c). Product release is thus fast for both 8-oxo-rGTP(syn) and -dGTP (syn)[24]. This is influenced by active site geometry, whereby the syn-conformation induces more strain in the $PP_i$ binding region, than the anti-conformation, precipitating $PP_i$ release, and contributing toward the improved efficiency and enhanced mutagenesis resulting from factors hastening the forward reaction (Fig. 6). Strikingly, $PP_i$ along with $Mn_p$ is retained for 16 h in the 8-oxo-dGTP(anti):$C_t$ product complex[24]. In the 16 h soak of the 8-oxo-rGTP(anti):$C_t$ product complex, density for $PP_i$ is reduced and any anomalous signal for $Mn_p$ is lacking due to its reduced occupancy (Supplementary Figs. 3c, 4a). However, product formation required a longer soak than for the 8-oxo-dGTP(anti):$C_t$ insertion. We observed 16 h to be the optimal time, as the occupancy of $Mn_p$ was reduced to undetectable and product formation was complete. These observations generally suggest that $Mn_p$ negatively influences the efficiency of the reverse reaction by stabilizing the product complex, improving efficiency of the overall forward (synthesis) reaction (Fig. 6). The decreased stability or lack of $Mn_p$ in the 8-oxo-rGTP(anti):$C_t$ insertion also demonstrates that the product metal is not required for DNA synthesis[28,29,49,50].

These observations suggest that the combination of O2´ and O8 places excess steric and electrostatic strain on the rigid active site, and the primer terminus responds through an altered conformation (see Fig. 2). Although pol μ lacks gross subdomain motions or template strand adjustments, repositioning of the palm domain and associated regions, including the truncated loop 2, accommodate 8-oxo-rGTP binding (Supplementary Fig. 2). Limited global conformational adjustments may therefore play a role in low-fidelity discrimination of 8-oxo-rGTP insertion by pol μ. Subdomain and template strand motions play key roles in enforcing discrimination by pols β[41] and λ[42], respectively. Pol μ lacks these features[6,8], where the rigid active site and bond formation appear to force substrates and catalytic moieties into similar positions, promoting base-pair stabilization and insertion of noncanonical nucleotides[24].

## Methods

**Protein expression and purification**. Truncated human pol μ was overexpressed in BL21-CodonPlus(DE3)-RIL cells (Invitrogen)[9]. Cells were harvested in Lysis buffer (25 mM Tris/HCl, pH 8.0 (25 °C), 5% glycerol, 500 mM NaCl, 1 mM DTT) and lysed by sonication on ice. Batch purification using Glutathione Sepharose CL-4B resin (GE Healthcare) was followed by size-exclusion chromatography in Gel-Filtration buffer (25 mM Tris/HCl, pH 8.0 (25 °C), 5% glycerol, 500 mM NaCl, 1 mM DTT, 1 mM EDTA). Pol μ was then dialyzed into Storage buffer (25 mM Tris/HCl, pH 8.0 (25 °C), 5% glycerol, 500 mM NaCl, 1 mM DTT), concentrated to 11 mg ml$^{-1}$, and stored at −80 °C.

**Gap-filling kinetic assays**. The 34-mer template oligonucleotide (3´-GACGTC-GACTACGCG X CATGCCTAGGG GCCCATG-5´ where X = A or C, Supplementary Table 8) was annealed to a 17-mer upstream and a 15-mer 5´-phosphorylated downstream oligonucleotide in a 1.2:1.2:1 ratio (template:

downstream: upstream) in Annealing buffer (10 mM Tris-HCl, pH 7.5, 1 mM EDTA). Pol μ (5 nM) was pre-incubated with 100 nM single-nucleotide-gapped DNA substrate and mixed with rGTP or 8-oxo-rGTP in Reaction buffer (50 mM Tris-HCl, pH 7.4 (37 °C), 10 mM MgCl$_2$ or 10 mM MnCl$_2$, 100 mM KCl, 10% glycerol, 100 μg ml$^{-1}$ bovine serum albumin, 1 mM dithiothreitol, 0.1 mM EDTA). Reactions were terminated by addition of 0.25 M EDTA and an equal volume of formamide dye. Products were separated on an 18% denaturing gel and quantified with a Typhoon phosphorimager. The apparent insertion rate ($k_{cat,app}$) and equilibrium Michaelis constants ($K_{m,app}$) were determined by fitting to the Michaelis–Menten equation.

DNA was purchased PAGE purified from Integrated DNA Technologies (Coralville, IA).

**Time-lapse crystallography**. Crystallography was performed by annealing a 9-mer template oligonucleotide (5´- CGGC X TACG-3´ where X = A or C, Supplementary Table 8) with a 4-mer upstream (5´-CGTA-3´) oligonucleotide and a 5´-phosphorylated downstream 4-mer (5´-pGCCG-3´) oligonucleotide in a 1:1:1 ratio in 100 mM Tris-HCl pH 7.5 (Supplementary Fig. 1). Pol μ-DNA binary complex crystals were transferred to a cryo-solution (15% ethylene glycol, 100 mM HEPES pH 7.5, 20% PEG4000, 5% glycerol, 50 mM NaCl, 2 mM 8-oxo-rGTP or 2 mM 8-oxo-dGTP, and 20 mM CaCl$_2$) for 120 min. Ground state ternary complex crystals were transferred to a cryo-solution (preceded by a pre-soak wash) containing 50 mM MgCl$_2$ or 50 mM MnCl$_2$ for varying times. Reactions were terminated by plunging the crystal into liquid nitrogen. DNA was purchased PAGE purified from Integrated DNA Technologies (Coralville, IA).

**Data collection and refinement**. Data collection was performed at the Advanced Photon Source (Argonne National Laboratory, Chicago, IL) on the ID22 beamline (Southeast Regional Collaborative Access Team, SER-CAT) using the Mar300HX or Eiger16M detectors at a wavelength of 1.00 Å. Data were also collected on an in-house Rigaku Saturn 944+ CCD detector mounted on a MiraMax-007HF rotating anode generator at a wavelength of 1.54 Å. Data were processed and scaled using the programs HKL2000[51] or HKL3000[52]. Initial models were determined using molecular replacement with a previously determined structure of Pol μ (PDB id 4M04[9]). Refinement was carried out using the PHENIX software package[53] and iterative model building was done using Coot[54]. All R$_{free}$ flags were taken from the starting model, partial catalysis models were generated with both the reactant and product species, and occupancy refinement was performed. Ramachandran analysis determined 100% of nonglycine residues lie in allowed regions and at least 97% in favored regions. All omit density maps were generated by deleting the regions of interest and performing simulated annealing. Unless otherwise indicated, simulated annealing omit ($F_c$–$F_c$) density maps are shown as a green mesh contoured at 3 σ, carve radius 2.0 Å. The figures were prepared in PyMol (Schrödinger).

**Molecular dynamics simulations**. Interaction free energies of nucleotide triphosphates were calculated using molecular dynamics (MD) simulations, according to the following. Missing atoms and protons were introduced into the initial structures (PDB ids, 6VFA in the unreacted conformation, and 6VFB in the reacted conformation) using the tleap module of Amber.18[55], counterions were added, and the systems were solvated in a box of water with the box boundary extending to 20 Å from the nearest peptide (or DNA) atom (resulting in 77602, 81521, and 83465 atoms in each simulation box). Prior to equilibration, all systems were subjected to (1) 500 ps of belly dynamics with fixed peptide, (2) minimization, (3) low temperature constant pressure dynamics at fixed protein to assure a reasonable starting density, (4) minimization, (5) step-wise heating MD at constant volume, and (6) constant volume simulation for 10 ns with a constraint force constant of 10 kcal mol$^{-1}$ applied only on backbone heavy atoms. The next 50 ns was used to step-wise reduce the contain force constant to 1.0 kcal mol$^{-1}$. Subsequently, unconstrained MD simulations were extended for 200 ns. Trajectories were calculated using the PMEMD module of Amber.18 with 1 fs time step. The amino acid parameters were selected from the SB14ff force field of Amber.18 and the DNA force field was parmbsc1. Using 100 configurations from each simulation selected at 1 ns intervals of the last 100 ns, interaction free energies of nucleotide triphosphates and residue interaction energies were estimated with the MMGBSA module of Amber.18, at the salt concentration of 150 mM.

**Reporting summary**. Further information on research design is available in the Nature Research Reporting Summary linked to this article.

## Data availability

Atomic coordinates and structure factors for the reported crystal structures have been deposited in the Protein Data Bank (PDB) under accession numbers: 6VEZ, 6VF0, 6VF1, 6VF2, 6VF3, 6VF4, 6VF5, 6VF6, 6VF7, 6VF8, 6VF9, 6VFA, 6VFB, and 6VFC. All data are available from the authors upon reasonable request.

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

## Acknowledgements

We wish to thank SER-CAT for assistance with data collection. Use of the advanced Photon Source was supported by the U.S. Department of Energy, Office of Science, Office of Basic Energy Sciences, under contract W-31-109-Eng-38. This research was supported by research project numbers Z01-ES050158 and Z01-ES050161 (S.H.W.), Z01-ES043010 (L.P.), and K99ES029572-01 (J.A.J.) in the intramural research program of the National Institutes of Health, NIEHS. Support was also provided by JSPS KAKENHI Grant Number 16K16195 (A.S.).

## Author contributions

J.A.J. designed the study and performed the crystallography; A.S. and D.D.S. performed the kinetics; L.P. performed the MD simulations; J.A.J., A.S., L.P., D.D.S., W.A.B., and S.H.W. analyzed the data; J.A.J. and S.H.W. wrote the paper; S.H.W. supervised the research.

## Funding

## Competing interests

The authors declare no competing interests.
