## [Peer Review File · Nature Communications]

REVIEWER COMMENTS

Reviewer #1 (Remarks to the Author):

This manuscript focuses on the ability of pol μ to discriminate against 8-oxo-rGTP and appears to be a companion paper to another manuscript that examines 8-oxodGTP incorporation. Interestingly, it appears that 8-oxo-rGTP incorporation is particularly problematic because the normal cellular repair systems are very inefficient in the recognition/repair of the oxidized ribonucleotide (compared to the deoxy). The data is largely structural, along with the addition of a kinetic experiment examining the catalytic efficiencies of the nucleotides with a templating C or A. The kinetic data seem reasonable, although in the Methods section the concentration of the polymerase used in the assays is not given. This definitely needs to be provided because it's not clear that steady-state conditions are being maintained during the assays (i.e., [DNA] \gg [pol]). The remainder of the manuscript describes time-lapse x-ray crystallography experiments of the 8-oxo-rGTP incorporation with Mn²⁺ or Mg²⁺. The results section is quite dense, but fairly easy to follow and the figures do a good job at showing what the authors intend (although obviously these 2-dimensional figures leave a lot to be desired). Overall, this manuscript is enjoyable to read and it describes high quality, interesting crystal structures of 8-oxo-rGTP incorporation. It should be of broad interest to the fields of DNA replication and repair, and enzymologists in general.

A few specific comments are below:

-Page 6 line 9. This sentence stating that Mn²⁺ might possibly be the physiological metal should have a reference to support it.

In Fig 3b, should it be Reactant (Palpha) instead of (Pgamma)?

Missing figure legend for Fig 3d

I think you should state the occupancy values from the extended data table 3 when discussing binding at reduced occupancy. For example, line 14 on page 12. It would be easier to have the value in parenthesis, rather than looking it up in the table.

Line 15, page 13.

Is "undamaged" the correct word here? This section is only discussing ribonucleotides, correct? If so, I'm not sure a ribonucleotide should be considered damaged.

Line 17, page 17. The term "released into solution" seems misleading to me. What exactly is meant by this?

Reviewer #2 (Remarks to the Author):

Jamsen and colleagues describe new structures for nucleotide incorporation by human pol μ in double strand break repair, which is supposed to differ from structures done previously in the same laboratory for DNA template/primer duplexes that are not involved in double strand break repair. However, this reviewer has a difficult time to find how these new structures differ from earlier ones and how they represent DNA template/primer duplexes involved in DNA double strand break repair. Authors need to show a diagram what the structure is supposed to look like in the experimental design, and how this DNA template/primer duplex is actually bound in the crystal structure before one focuses on description of the catalytic site, which seems to be the bulk of the study.

To my best understanding, 9-mer DNA template (5'-CCGCATACG-3') is base paired with a 4-mer (5'-CGTA-3') oligonucleotide and another 4-mer (5'-pGCCG-3') so that there is a gap in the middle with a templating A nucleotide used the DNA primer/template duplex design in their time-resolved crystallography. How does this DNA primer/template duplex resemble the said DNA double strand break? In crystallization, it was stated 1mM dNTP was included. Which dNTP was it? Was it dTTP? Why was not 8-oxo-rGTP described in the methods and experimental procedures? If not, where did it come from the Results section? The results section described the structures 8-oxo-rGTP base

paired with both At and Ct templating nucleotide, neither corresponding DNA sequence was described in the methods.

For this reason, this reviewer does not understand what has actually been done in this study before trying to understand what has been observed. This manuscript requires extensive revision before one can begin to review it.

Page 3 Line 8: What does Tdt stand for?

Page 39-40: There is no figure legend for Fig. 3d.

Response to Reviewer Comments

We thank the Reviewers for their careful reading, constructive remarks, and insightful comments. A detailed response to each comment is given below. We believe the revised manuscript has been substantially improved. We hope it is now suitable for publication.

Page and line numbers refer to the revised manuscript and all changes in the main text are highlighted in yellow.

REVIEWER COMMENTS

REVIEWER #1

Comment (1): The kinetic data seem reasonable, although in the Methods section the concentration of the polymerase used in the assays is not given. This definitely needs to be provided because it's not clear that steady-state conditions are being maintained during the assays (i.e, [DNA] >> [pol]).

Author Response: The polymerase concentration was 5 nM. This has been added to the Methods section (p. 23, line 17).

Comment (2): The remainder of the manuscript describes time-lapse x-ray crystallography experiments of the 8-oxo-rGTP incorporation with Mn²⁺ or Mg²⁺. The results section is quite dense, but fairly easy to follow and the figures do a good job at showing what the authors intend (although obviously these 2-dimensional figures leave a lot to be desired). Overall, this manuscript is enjoyable to read and it describes high quality, interesting crystal structures of 8-oxo-rGTP incorporation. It should be of broad interest to the fields of DNA replication and repair, and enzymologists in general.

Author Response: We thank the Reviewer for these positive comments.

Specific Comment (1): Page 6 line 9. This sentence stating that Mn²⁺ might possibly be the physiological metal should have a reference to support it.

Author Response: The references were added to the revised manuscript (p. 6, line 10).

Specific Comment (2): In Fig 3b, should it be Reactant (Palpha) instead of (Pgamma)?

Author Response: The label refers to Pgamma of 8-oxo-rGTP bound opposite template C in an unreactive (or "reverse") orientation in the active site (Fig. 3a, Supplementary Figs. 3a-b).

Simulated annealing omit ($F_o - F_c$) density at 3σ for the triphosphate was strong but density for the base and sugar was absent in the Ca^{2+} -ground state 8-oxo-rGTP: $\text{C}_i(\textit{anti})$ ternary complex (Fig. 2b). A short soak of these crystals in a Mn^{2+} containing cryo-solution revealed additional density for the base and sugar (Figs. 2c-d, Supplementary Figs. 2d). Density for the incorporated nucleotide was observed in the reacted conformation in the product state (Fig. 3c). In all cases, evidence for the conformations of the primer terminus was consistent and reproducible (Figs. 2b, 2d, 3a, Supplementary Fig. 3b).

We modeled the major 8-oxo-rGTP conformation in the 8-oxo-rGTP: $\text{C}_i(\textit{anti})$ reaction state ternary complex as bound in an unreactive reverse orientation in the active site. This is consistent with the Mn^{2+} -ground state 8-oxo-rGTP: $\text{C}_i(\textit{anti})$ ternary complex (Fig. 2d). The presence of additional minor conformations cannot be excluded. The structure at this timepoint was highly reproducible.

Specific Comment (3): Missing figure legend for Fig 3d

Author Response: The figure legend has now been added to the manuscript (p. 42, lines 4-9).

Specific Comment (4): I think you should state the occupancy values from the extended data table 3 when discussing binding at reduced occupancy. For example, line 14 on page 12. It would be easier to have the value in parenthesis, rather than looking it up in the table.

Author Response: The occupancies are now stated in the main text (p. 13, line 17; p. 13, line 19).

Specific Comment (5): Line 15, page 13. Is “undamaged” the correct word here? This section is only discussing ribonucleotides, correct? If so, I’m not sure a ribonucleotide should be considered damaged.

Author Response: This section discusses the Results in the context of ribonucleotide discrimination by pol mu. “undamaged” was replaced with “undamaged ribonucleotide” (p. 15, line 2). “Undamaged” was removed from a following sentence (p. 15, line 5).

Specific Comment (6): Line 17, page 17. The term “released into solution” seems misleading to me. What exactly is meant by this?

Author Response: The term “released into solution” is used in connection with the so-called substrate channeling hypothesis whereby metabolic reaction cascades compartmentalize metabolic intermediates in order to prevent the degradation and possible release of unstable species or byproducts of these reactions. This is thought to improve the efficiency of metabolic pathways.

Similarly, repair intermediates from one step of the DNA repair pathway are handed off to the next step or enzyme in order to improve the efficiency of repair and prevent the release of unstable repair intermediates. Upon failure of repair, a DSB repair intermediate could impair genomic integrity if “released into solution” and signal apoptosis, or have other adverse consequences. This section was modified (p. 19, lines 17-18).

REVIEWER #2

Comment (1): However, this reviewer has a difficult time to find how these new structures differ from earlier ones and how they represent DNA template/primer duplexes involved in DNA double strand break repair. Authors need to show a diagram what the structure is supposed to look like in the experimental design, and how this DNA template/primer duplex is actually bound in the crystal structure before one focuses on description of the catalytic site, which seems to be the bulk of the study.

Author Response: A diagram has been included as Supplementary Fig. 1 of the revised manuscript. A binary pol mu-primer/template duplex (PDB id 4LZG) structure of a double strand break (DSB) repair intermediate with a single nucleotide gap was reported by Moon et al. (2014) *Nat Struc Mol Biol* 21(3): 253-260. A binary pol mu-primer/template duplex structure of a DSB repair intermediate with a single nucleotide gap in one strand and a nick in the opposing strand was reported by Kaminski et al. (2020) *Nat commun* 11(1): 4784. We refer the Reviewer to these publications for more information on the binary complex. The ternary pol mu-primer/template duplex with bound 8-oxo-rGTP is shown in Supplementary Figs. 2a-b.

Comment (2): To my best understanding, 9-mer DNA template (5'-CCGCATACG-3') is base paired with a 4-mer (5'-CGTA-3') oligonucleotide and another 4-mer (5'-pGCCG-3') so that there is a gap in the middle with a templating A nucleotide used the DNA primer/template duplex design in their time-resolved crystallography. How does this DNA primer/template duplex resemble the said DNA double strand break?

Author Response: A DNA double strand break (DSB) contains breaks in both strands of the double helix. A single nucleotide gap DSB repair intermediate is processed in order to complete repair of the DSB, as shown in Supplementary Fig. 1. Our approach is strongly supported by a recent study by Kaminski et al. (2020) in *Nature communications*, where the authors determined that nucleotide insertion by pol mu on a DSB repair intermediate with a single nucleotide gap in one strand and a nick in the opposing strand (intermediate 2 in Supplementary Fig. 1 with nick in the template (grey) strand) proceeds identically to that on a single nucleotide gap DSB repair intermediate. Additionally, pol mu ribonucleotide insertion was recently shown to be a requirement for efficient non-homologous end-joining DSB repair in vivo (Pryor et al. (2018) *Science*

361: 1126-1129), and 8-oxo-rGTP is the major cellular oxidized ribonucleotide in the cellular nucleotide pool.

Comment (3): In crystallization, it was stated 1mM dNTP was included. Which dNTP was it? Was it dTTP? Why was not 8-oxo-rGTP described in the methods and experimental procedures? If not, where did it come from the Results section?

Author Response: We revised the Materials and Methods to address this discrepancy. 1mM dNTP has been replaced with 2 mM 8-oxo-rGTP or 2 mM 8-oxo-dGTP in the revised manuscript (p. 24, line 15).

Comment (4): The results section described the structures 8-oxo-rGTP base paired with both At and Ct templating nucleotide, neither corresponding DNA sequence was described in the methods. For this reason, this reviewer does not understand what has actually been done in this study before trying to understand what has been observed. This manuscript requires extensive revision before one can begin to review it.

Author Response: The DNA template sequence used in crystallization was 5´-CGGC X TACG-3´, where X = A or C) (p. 24, lines 9-10). We clarified the Methods section to address these discrepancies. We appreciate the assistance in pointing them out and hope the manuscript is now more readable for the Reviewer.

Comment (5): Page 3 Line 8: What does Tdt stand for?

Author Response: Tdt denotes Terminal deoxynucleotidyl transferase in short hand notation. This term has been defined in the revised manuscript (p. 3, line 9).

Comment (6): Page 39-40: There is no figure legend for Fig. 3d.

Author Response: The figure legend has now been re-added to the manuscript (p. 42, lines 4-9).

REVIEWERS' COMMENTS

Reviewer #1 (Remarks to the Author):

I am satisfied with the authors' responses and the manuscript revisions. All of my question and concerns have been addressed.

Reviewer #2 (Remarks to the Author):

Jamsen et al. (ms # 256946_1) report results of biochemical and crystallographic experiments with molecular dynamics simulation with well documented evidence. The authors objectively discuss what is observed in regard to the product metal ion (for example, in Figure 6). I recommend the acceptance of this paper for publication pending minor revision after addressing the following issue.

On page 21 lines 7-12: Strikingly, PPI along with Mnp, is retained at almost full occupancy for 16 h in the 8-oxo-dGTP(anti):Ct product complex, however, any reverse reaction is not apparent in the crystal. Density for PPI is reduced and any anomalous signal for Mnp is lacking in the longer soak of the 8-oxo-rGTP(anti):Ct product complex (Supplementary Figs. 3c, 4a), although on a much longer than for 8-oxo-dGTP(anti):Ct.

I recommend that authors to cite Jamsen et al. (accepted) in the first statement because the dGTP(anti):Ct structure is not part of this study. I am not certain whether authors can conclude that any reverse reaction is not apparent in the crystal because what is observed in the crystal is partial equilibrium of the forward and reverse reactions as long as PPI:Mnp remains bound. With increasing time, PPI:Mnp is dissociated slowly and thus shifts the equilibrium.

The second sentence is convoluted with two different ideas. Please revise as follows, for example. For the 8-oxo-rGTP(anti):Ct product complex with 16 h soaking, density for PPI is reduced and there is no detected anomalous signal for Mnp due to its reduced occupancy. However, to see this Mnp ion in the 8-oxo-rGTP(anti):Ct complex, the soaking time required (16 h) is longer than that of the dGTP(anti):Ct structure. We observed that 16 h remain to be the optimal time because at any time point in the 8-oxo-rGTP(anti):Ct complex the occupancy of Mnp is reduced to undetectable.

On page 21 lines 15-17: The decreased stability or lack of Mnp in the 8-oxo-rGTP(anti):Ct insertion also demonstrates that the product metal is not required for DNA synthesis.

Jamsen et al. are not the first investigators to question whether the product metal ion is involved in DNA synthesis in general since it was first reported, for example, Wang and Smithline (2019, Protein Science 28, 439-447), to which citation should be considered to include. Evidence provided by Jamsen et al is very convincing. In general, what is observed is partial equilibrium or conditional equilibrium. At time zero, the reaction is dominated by the forward reaction when the product concentration is zero. With increasing time, the contribution of the reverse reaction increases with the increasing product. After a certain time point, it decreases after PPI:Mnp is irreversibly dissociated.

Response to Reviewer Comments

We thank the Reviewers for their careful reading, constructive remarks, and insightful comments. A detailed response to each comment is given below. We believe the revised manuscript has been substantially improved. We hope it is now suitable for publication.

Page and line numbers refer to the revised manuscript and all changes in the main text are highlighted in yellow.

REVIEWER COMMENTS

Reviewer #1 (Remarks to the Author):

I am satisfied with the authors' responses and the manuscript revisions. All of my question and concerns have been addressed.

Authors: We thank the reviewer for their time and effort.

Reviewer #2 (Remarks to the Author):

Jamsen et al. (ms # 256946_1) report results of biochemical and crystallographic experiments with molecular dynamics simulation with well documented evidence. The authors objectively discuss what is observed in regard to the product metal ion (for example, in Figure 6). I recommend the acceptance of this paper for publication pending minor revision after addressing the following issue.

On page 21 lines 7-12: Strikingly, PPi along with Mnp, is retained at almost full occupancy for 16 h in the 8-oxo-dGTP(anti):Ct product complex, however, any reverse reaction is not apparent in the crystal. Density for PPi is reduced and any anomalous signal for Mnp is lacking in the longer soak of the 8-oxo-rGTP(anti):Ct product complex (Supplementary Figs. 3c, 4a), although on a much longer than for 8-oxo-dGTP(anti):Ct.

I recommend that authors to cite Jamsen et al. (accepted) in the first statement because the dGTP(anti):Ct structure is not part of this study. I am not certain whether authors can conclude that any reverse reaction is not apparent in the crystal because what is observed in the crystal is partial equilibrium of the forward and reverse reactions as long as PPi:Mnp remains bound. With increasing time, PPi:Mnp is dissociated slowly and thus shifts the equilibrium.

The second sentence is convoluted with two different ideas. Please revise as follows, for example. For the 8-oxo-rGTP(anti):Ct product complex with 16 h soaking, density for PPi is reduced and there is no detected anomalous signal for Mnp due to its reduced occupancy. However, to see this Mnp ion in the 8-oxo-rGTP(anti):Ct complex, the soaking time required (16 h) is longer than that of the dGTP(anti):Ct structure. We observed that 16 h remain to be the optimal time because at any time point in the 8-oxo-rGTP(anti):Ct complex the occupancy of Mnp is reduced to undetectable.

Authors: (p. 21 lines 6-11) We modified the sentence as suggested by the Reviewer.

On page 21 lines 15-17: The decreased stability or lack of Mnp in the 8-oxo-rGTP(anti):Ct insertion also demonstrates that the product metal is not required for DNA synthesis.

Jamsen et al. are not the first investigators to question whether the product metal ion is involved in DNA synthesis in general since it was first reported, for example, Wang and Smithline (2019, Protein Science 28, 439-447), to which citation should be considered to include. Evidence provided by Jamsen et al is very convincing. In general, what is observed is partial equilibrium or conditional equilibrium. At time zero, the reaction is dominated by the forward reaction when the product concentration is zero. With increasing time, the contribution of the reverse reaction increases with the increasing product. After a certain time point, it decreases after PPI:Mnp is irreversibly dissociated.

Authors: (p. 21, line 16) We added the citation recommended by the Reviewer. We modified Figure 6 to accommodate the Reviewer's comments.